# Millimetre-long transport of photogenerated carriers in topological insulators

Yasen Hou[1], Rui Wang[2], Rui Xiao[1], Luke McClintock[1], Henry Clark Travaglini[1], John Paulus Francia[1], Harry Fetsch[3], Onur Erten[4], Sergey Y. Savrasov[1], Baigeng Wang[5], Antonio Rossi, Inna Vishik [1], Eli Rotenberg [6] & Dong Yu [1]*

Excitons are spin integer particles that are predicted to condense into a coherent quantum state at sufficiently low temperature. Here by using photocurrent imaging we report experimental evidence of formation and efficient transport of non-equilibrium excitons in $Bi_{2-x}Sb_xSe_3$ nanoribbons. The photocurrent distributions are independent of electric field, indicating that photoexcited electrons and holes form excitons. Remarkably, these excitons can transport over hundreds of micrometers along the topological insulator (TI) nanoribbons before recombination at up to 40 K. The macroscopic transport distance, combined with short carrier lifetime obtained from transient photocurrent measurements, indicates an exciton diffusion coefficient at least $36\,m^2\,s^{-1}$, which corresponds to a mobility of $6 \times 10^4\,m^2\,V^{-1}\,s^{-1}$ at 7 K and is four order of magnitude higher than the value reported for free carriers in TIs. The observation of highly dissipationless exciton transport implies the formation of superfluid-like exciton condensate at the surface of TIs.

[1] Department of Physics, University of California, Davis, CA 95616, USA. [2] Department of Physics and Astronomy, Shanghai Jiao Tong University, 200240 Shanghai, China. [3] Department of Physics, Harvey Mudd College, Claremont, CA 91711, USA. [4] Department of Physics, Arizona State University, Tempe, AZ 85281, USA. [5] Department of Physics, Nanjing University, 210008 Jiangsu, China. [6] Advanced Light Source, Lawrence Berkeley National Laboratory, Berkeley, CA 94720, USA. *email: yu@physics.ucdavis.edu

Avariety of systems, including double quantum wells[1–4], microcavities[5], graphene[6,7] and transition metal dichalcogenides[8], have shown signatures of exciton condensation. Dirac materials such as graphene and topological insulators (TIs) with strong Coulomb attraction and vanishing effective mass emerge as a new promising platform for achieving exciton condensate potentially at room temperature[9–11]. The gapless TI surface state is protected against backscattering and has a linear energy dispersion with massless fermions. Although free-fermions have been extensively studied in TIs, much less work is carried out to understand interacting systems[12,13], where electron–electron interaction may lead to emerging quasi-particles. Photoexcited electrons and holes in TIs relax to the surface Dirac cones on sub-picosecond (ps) timescales, while further carrier recombination can be much slower, ranging from a few ps to over 400 ps[14–19]. This long-lived population inversion allows electrons and holes in the transient state to form pairs (Fig. 1a). Because of the small effective mass, excitons in Dirac materials are expected to have long de Broglie wavelength and high transition temperatures $(T_c)$[9,10]. The figure of merit for exciton formation in materials is $\alpha = \frac{E_C}{E_K}$, where $E_C$ is the Coulomb energy and $E_K$ is the electron kinetic energy. The linear dispersion of the TI surface state results in $\alpha = e^2/\epsilon \hbar v_F$, where $e$ is the electron charge, $\epsilon$ is the dielectric constant of the material and $v_F$ is the Fermi velocity[20]. The two-dimensional (2D) surface state of a three-dimensional (3D) TI, with a single non-degenerate Dirac cone, relatively low $v_F$ (compared to graphene) and reduced $\epsilon$ at surface, has been theoretically identified as a promising candidate for realizing high-$T_c$ exciton condensates[10,11]. In addition, the topological nature of the band structure may create exotic spin texture to the excitonic quantum state. The spin-momentum locking demands that the ground state of excitons must be a spin-triplet $p$-wave, which spontaneously breaks time reversal symmetry[21].

Previous experimental evidence of exciton condensation in gapped semiconductors has been obtained from spatially resolved photoluminescence (PL) measurements, where PL images exhibit macroscopically ordered patterns[2], or PL peak intensity sharply increases with reduced peak widths at lower temperature[3,5]. More recently, exciton formation has been experimentally demonstrated in both graphene[6,7,22] and TIs[23]. Evidence of superfluidic excitons has also been obtained by quantum Hall drag in bilayer graphene[6,7]. Photocurrent imaging is a powerful experimental technique that can be applied to visualize the transport of locally photoexcited charge carriers[24,25]. Compared to spatially resolved PL, it does not require materials to have strong light emission and is hence ideal to study TIs. Previous photocurrent studies of TIs have largely been on degenerately $n$-doped TIs, where photocurrent is weak with an external quantum efficiency (EQE) of <1% and photocurrent decays rapidly as the local photoexcitation moves away from the electrical contacts to the TIs[26–28].

Here, we apply scanning photocurrent microscopy (SPCM) in intrinsic 3D TIs to provide evidence on the formation of excitons and their transport of macroscopic distance.

## Results

**Non-local photocurrent.** $Bi_{2-x}Sb_xSe_3$ nanoribbons were grown by chemical vapour deposition (CVD)[29], with $x = 0.38$ determined from energy-dispersive X-ray spectra. Sb doping significantly suppresses bulk conduction as evidenced by field-effect characteristics (detailed later). The experimental setup is shown in Fig. 1a, where a nanoribbon is electrically connected by two metal contacts (Fig. 1b) and is locally excited by a focused laser. As the laser beam is raster scanned on the device substrate, the photo-induced current is measured as a function of laser position

and plotted into a 2D map (Fig. 1c). At room temperature, photocurrent is only observed when the laser is focused close to the contacts, caused by photo-thermoelectric effects. As the temperature is reduced, the photocurrent becomes much stronger and its direction is reversed. Importantly, the photocurrent is highly non-local and can be observed even when the laser is focused far outside the channel between the contacts. Strikingly, below 40 K the photocurrent barely decays even when the photoexcitation position is more than 200 μm away from the contact (Fig. 1c, d). The photocurrent decay length $(L_d)$ at various temperatures is determined by fitting photocurrent distributions with a hyperbolic function $I(x_0) = A \cosh\left(\frac{x_0 - L}{L_d}\right)$, where $x_0$ is the excitation position and $L$ is the length of the nanoribbon outside the channel (more details on $L_d$ extraction and error analysis in Supplementary Notes 1 and 2). Remarkably, $L_d$ is below 3 μm at 200 K but increases to $0.9 \pm 0.3$ mm at 7 K, accompanied by an internal quantum efficiency (IQE) as high as 60% (Fig. 1e). The observed non-local photocurrent is robust and highly reproducible in more than 10 devices measured to date.

The out-of-channel nanoribbon segment is electric field free and photoexcited carriers are expected to diffuse in this region. Photocurrent distribution in normal semiconductors free of external electric field is understood by the diffusion of minority free carriers[24,25]. In this model, photocurrent decays exponentially with a characteristic length of $L_d = \sqrt{D\tau}$, where $D$ is the diffusion coefficient and $\tau$ is the lifetime of minority carriers. Transient photocurrent measurements showed $\tau = 15 \pm 5$ ns in our samples (Supplementary Fig. 6). Limited by the bandwidth of electronics, this value should be treated as an upper limit of the actual lifetime. Taking $\tau = 20$ ns and $L_d = 0.9$ mm, we estimate a lower limit of mobility $\mu = eL_d^2/\tau k_B T \approx 6 \times 10^4$ m² V⁻¹ s⁻¹ at 7 K. This is 6 orders of magnitude higher than the field-effect mobility determined in our devices $(\mu = 0.037$ m² V⁻¹ s⁻¹), and 4 orders of magnitude higher than the highest reported mobility in 3D TIs $(\mu \sim 1$ m² V⁻¹ s⁻¹)[30–32]. Note that though the electron backscattering at the surface of a 3D TI is forbidden, scattering into other angles is possible[33,34], resulting in finite carrier mobility. Therefore, free carrier diffusion does not explain the observation. Other photocurrent generation mechanisms, such as thermoelectric, photo-Dember effects, inhomogeneous doping and photo-recycling, are also excluded (Supplementary Note 3).

**Field-independent photocurrent distributions.** To understand this unusual behaviour, we performed SPCM as a function of source–drain bias $(V_{SD})$ at 7 K and found that the in-channel photocurrent profiles remain largely independent of $V_{SD}$ (Fig. 2a, c). This is striking as $L_d$ in normal semiconductors is expected to strongly depend on electric field, as experimentally demonstrated previously[35–37]. Free charge carriers move faster along the electric force, and slower against, leading to longer photocurrent decay near one contact and shorter near the other, as shown in Fig. 2d. Quantitatively, in this free carriers model[38], $L_d = \frac{2L_{diff}^2}{\sqrt{L_{drift}^2 + 4L_{diff}^2} \mp L_{drift}}$, where $L_{diff} = \sqrt{D\tau}$, $L_{drift} = \mu\tau E$ and the signs indicate $L_d$ measured at opposite electrodes. When $E$ increases above a threshold $E_c = \frac{k_B T}{eL_{diff}}$, $L_d$ becomes drift dominated and strongly depends on $E$. As shown in Fig. 2b, we applied $E$ up to 20 times of $E_c$, but the measured $L_d$ values remained largely constant. At maximum applied field, the measured $L_d$ value is 20 times lower than that predicted from the free carrier model. This discrepancy from the free carrier model indicates the formation of excitons. The motion of these charge neutral particles is not affected by external electric field, resulting in a $E$-independent $L_d$ (Fig. 2b, d). Note that the applied electric field here is still much lower than that

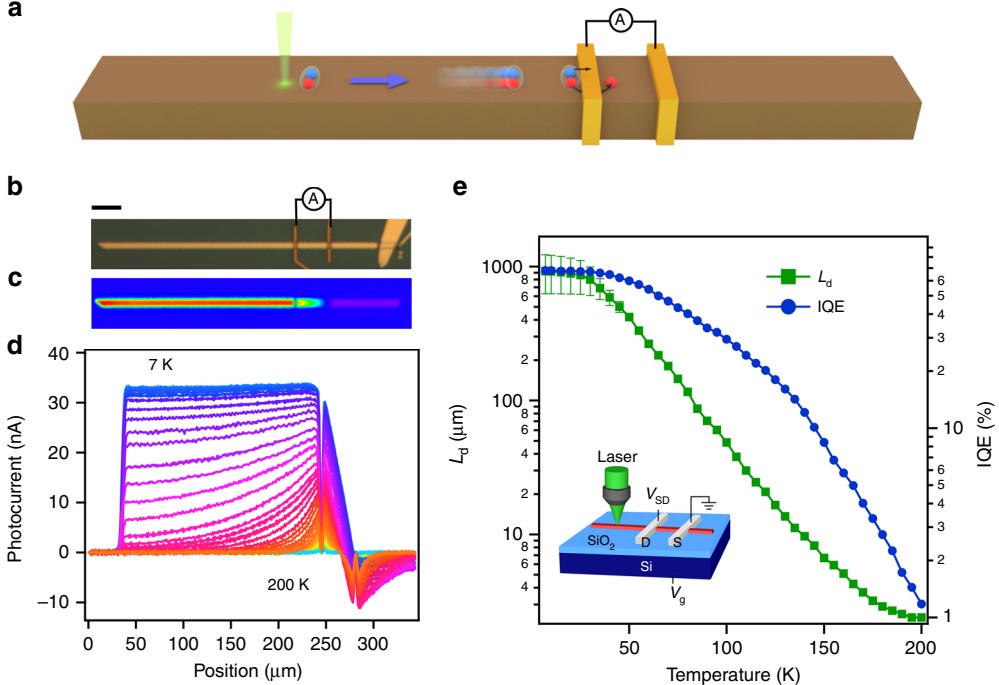

**Fig. 1 Non-local photocurrent generation in a TI nanoribbon. a** Schematic of exciton transport in TIs. Electrons and holes, denoted by blue and red balls, respectively, are bound and travel ballistically at TI surface until being separated at the metal contact. **b** Optical image of a Sb-doped $Bi_2Se_3$ nanoribbon ($305 \times 6.5 \times 0.13\ \mu m^3$) in contact with Cr/Au electrodes. The far right end of the nanoribbon is in contact with another TI nanoplate. The scale bar denotes 30 μm. **c** A photocurrent map collected by scanning a focused laser at normal incidence at 7 K and zero source–drain and gate biases. Laser power is 166 nW. **d** Photocurrent distributions along the nanoribbon axis at various temperatures. **e** $L_d$ and IQE as a function of temperature. IQE (electron collected per absorbed photon) is calculated from the photocurrent and laser power considering a reflectance of 30%. The uncertainty of $L_d$ becomes large at low temperature because $L_d$ is several times larger than $L$. Inset: SPCM setup.

needed to separate excitons as estimated in Supplementary Note 4. In addition, photoexcitation at low temperature mainly produce excitons, but with a small portion of free carriers presumably because of thermal activation, as evidenced by the small shift of the photocurrent baseline with $V_{SD}$ (Supplementary Fig. 8).

**Effects of Sb doping, gate, excitation wavelength and intensity.** Long $L_d$ is only observed in Sb-doped $Bi_2Se_3$ samples, in which the Fermi level ($E_F$) is close to the Dirac point evidenced by the ambipolar gate dependence (Fig. 3d) and angle-resolved photoemission spectroscopy (ARPES)[39]. Micro-ARPES spectra of these nanostructures have demonstrated clear Dirac cones and indicated that the samples are slightly *n*-doped relative to the Dirac point, but with $E_F$ below the bulk conduction band (Supplementary Fig. 2). $L_d$ in samples with low Sb doping is shorter than that with more Sb (Supplementary Fig. 9). In pure $Bi_2Se_3$ that is degenerately *n*-doped, photocurrent with much lower magnitude is observed solely near the contacts (Fig. 3a). This explains why non-local photocurrent has not been reported in TIs, though photocurrent mapping in TIs has been studied in previous work[26–28]. Consistent with the doping-dependent photocurrent, we also found that $L_d$ and IQE can be greatly modulated by gate voltage ($V_g$). Photocurrent first increases slightly at negative $V_g$ when $E_F$ is lowered closer to the Dirac point. But as $V_g$ becomes more negative and tunes the TI from *n*-type to *p*-type (Fig. 3d), both $L_d$ and IQE drop sharply (Fig. 4a, d).

The photocurrent distributions are also measured as a function of light polarization, light intensity and excitation wavelength. Both circularly and linearly polarized laser beams are applied, but the resulting photocurrent distributions are independent of the polarization due to the normal incidence of the laser used in this

work[28]. $L_d$ is found to decrease at higher laser intensity (Fig. 4b, e) but is independent of excitation wavelength in a wide range of 500–1700 nm (Fig. 4c, f). The latter rules out the possibility of surface plasmon polariton (SPP)[40] since the SPP propagation length is expected to be wavelength dependent[41,42]. The normal incidence configuration with light injection from free space also unlikely creates surface plasmon due to momentum mismatch. Surface plasmons are expected to exist at high electron density[40], but we only observed long photocurrent decay lengths in intrinsic samples. In addition, the wavelength independent $L_d$ confirms that the second Dirac cone 1.5 eV above the conduction band edge[43] is not involved in the exciton formation.

## Discussion

One possible way to understand the observed highly efficient carrier transport at low temperature is the formation of superfluid-like exciton condensate in TIs. Different from free carriers, which suffer from scattering, excitons are bosons and can condense into a coherent quantum state at low temperature. In this picture, the photoexcitation mainly generate charge carriers in the bulk of the TI material. Then, these photoexcited carriers undergo a fast relaxation process within a few picoseconds and relax to the surface states where recombination is much slower up to hundreds of picoseconds. The electrons and holes at the surface form excitons at sufficiently low temperature. The exciton formation opens many-body energy gaps at the surface states similar to the energy gap associated with Cooper pairs in superconductors[10,11]. The excitons propagate across the TI surface ballistically over hundreds of micrometres. The flow of excitons does not generate an electrical current because excitons are charge neutral. However, as excitons reach the metal-TI contact, they are separated and create photocurrent. $Bi_2Se_3$ makes

 **3**

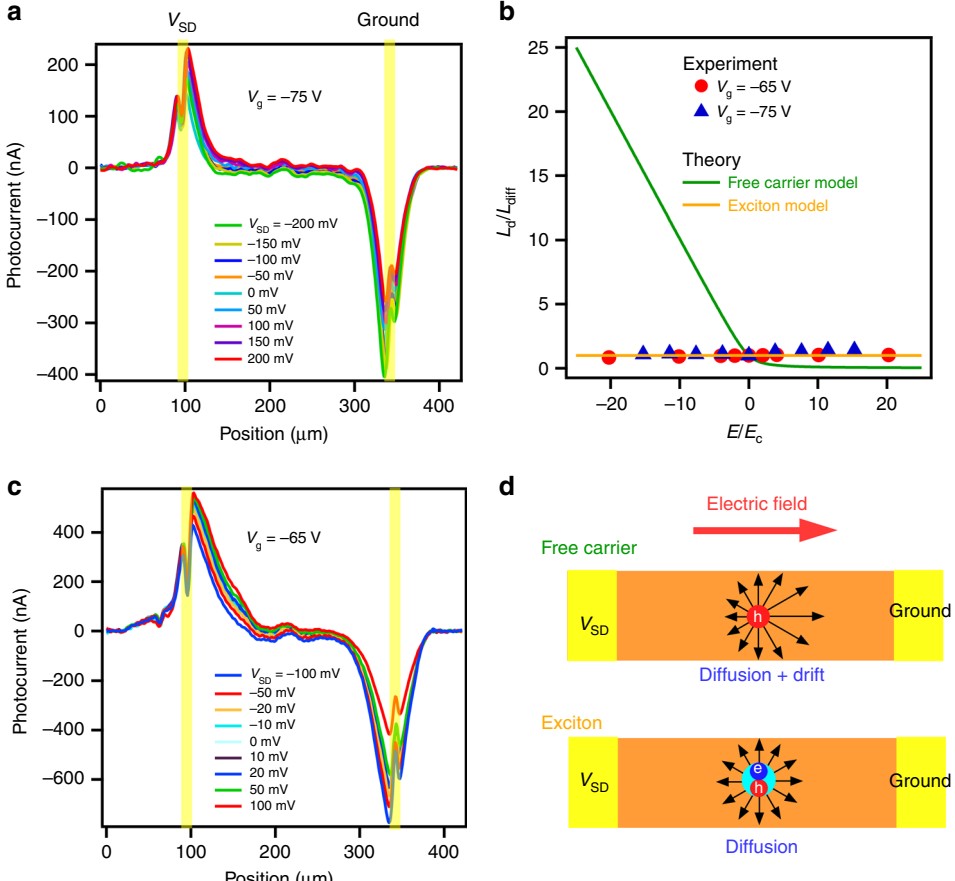

**Fig. 2 Electric field-independent photocurrent profiles at 7 K.** The dark current induced by $V_{SD}$ is subtracted from the total current. A gate voltage is applied to shorten $L_d$ in order to observe possible electric field induced changes. **a**, **c** Photocurrent as a function of laser excitation position along the nanoribbon axis at various $V_{SD}$ and $V_g = -75$ V ($L_{diff} = 11.4\,\mu m$) and $-65$ V ($L_{diff} = 30.1\,\mu m$), respectively. Laser power is $20\,\mu W$. Vertical yellow lines indicate the contacts. **b** $L_d$ near the left electrode extracted from **a**, **c** as a function of electric field $E$. $L_d$ is normalized by diffusion length $L_{diff}$ measured at $V_{SD} = 0$ V. $E$ is normalized by critical electric field $E_c$. Green and orange solid lines represent the photocurrent decay lengths predicted by the free carrier model[38] and exciton model, respectively. The field-independent $L_d$ indicates carriers are bound into charge neutral excitons. **d** Schematic showing that motion of free carriers is affected by external electric field, while excitons are not.

Ohmic contact to metals but strong band bending of hundreds of meV at the junction[44] facilitates efficient charge transfer (see band diagram and equivalent circuit model in Supplementary Fig. 3). The high IQE value at low temperature indicates that a large fraction of photoexcited carriers condense in the superfluid state.

This exciton condensate model is consistent with the sensitive dependence of photocurrent distributions on temperature, doping, gate and intensity. Both $L_d$ and IQE increase rapidly by orders of magnitude as temperature decreases and saturate below 40 K (Fig. 1e), which is consistent with the expectation of strong temperature dependence of Bose–Einstein condensate. This saturation temperature corresponds to $T_c = 40$ K, significantly higher than most of the previous reports of exciton condensation[1–6]. Long $L_d$ is only observed in intrinsic TIs when $E_F$ is close to the Dirac point, as shown by doping and gate effects. The rapid drop of $L_d$ as $V_g$ becomes more negative (Fig. 4a, d) is likely because of faster carrier recombination caused by the mixing of surface states and bulk valence band in $p$-type TIs, as the Dirac point is close to the bulk valence band in Bi$_2$Se$_3$. Furthermore, the strong dependence of $L_d$ on $E_F$ indicates that excitons are at the surface of TIs. While the carrier lifetime at the TI surface sensitively depends on $E_F$ and is over 400 ps in intrinsic samples, the lifetime in the bulk is always short in the order of

picoseconds[14–18]. As a result, excitons in the bulk are required to travel at a speed 2 orders of magnitude higher than the Fermi velocity in order to propagate across the 200-μm nanoribbon within this lifetime, which is highly unlikely. Finally, the strong light intensity dependence indicates that stronger screening at high intensity makes exciton formation more difficult. The theoretically estimated $T_c$ as a function of excitation power (Supplementary Note 4) is in good agreement with the experimental observation (Supplementary Figs. 10 and 11).

Different theories suggest that exciton condensates can be either an insulating[45,46] or a superfluid state[9,47,48]. Experimentally, signatures of both excitonic insulator[12] and excitonic superfluid[3,4,6,7] have been reported. The observed highly dissipationless transport of photogenerated carriers in TIs provides strong evidence supporting superfluidity. As pointed out in reference[48], superfluidity in He$^4$ or superconductors can be distinct from that in exciton condensate, where the former is via mass flow and the later is via energy flow. It is interesting to note that mass flow is not necessary for the observed long photocurrent decay. The energy flow from the photoexcitation point to electrical contact can also result in non-local photocurrent. Furthermore, Fig. 1e shows that $L_d$ decreases gradually when temperature is increased above $T_c$, indicating that the phase transition is not sharp. This can be understood by considering the

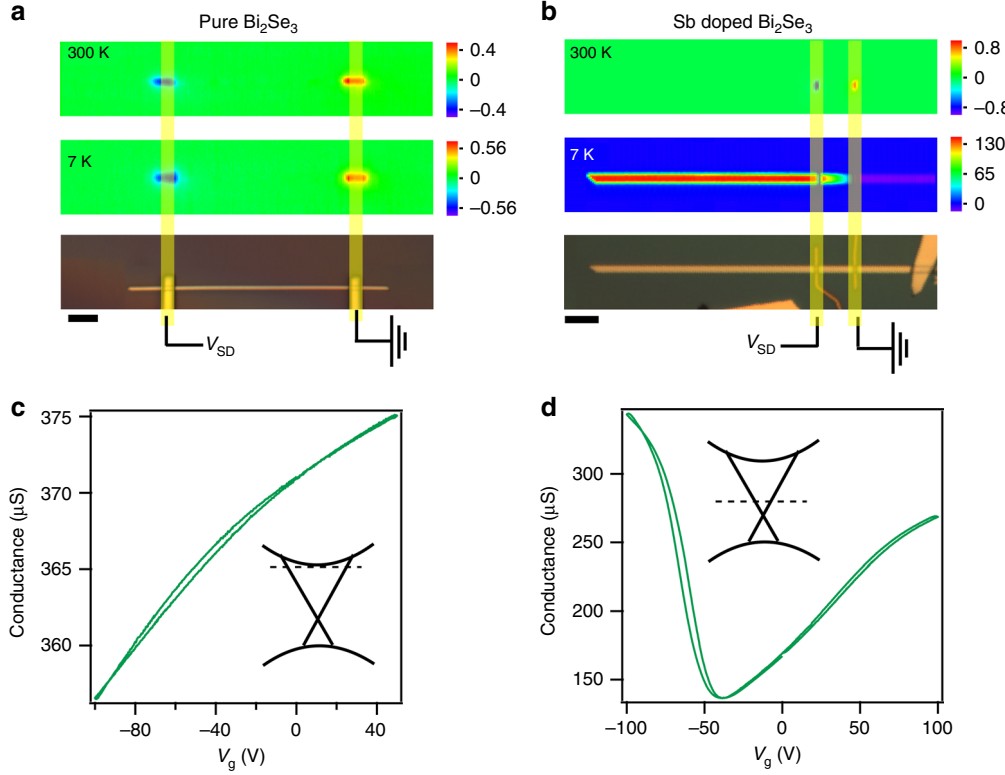

**Fig. 3 Doping-dependent photocurrent profiles. a, b** Photocurrent and optical images, where vertical yellow lines indicate the contacts. Colour scales are current in nanoampere. Laser power is 723 nW. **c, d** Gate-dependent conductance measured in the dark at 7 K. Insets: band diagrams showing $E_F$ positions. **a, c** are for pure $Bi_2Se_3$, where $E_F$ is close to the conduction band. Field-effect mobility and electron concentration are estimated to be $\mu = 329\ cm^2\ V^{-1}\ s^{-1}$, $n = 3.25 \times 10^{18}\ cm^{-3}$. The photocurrent is only observed when excitation is close to the contacts. **b, d** Sb doping lowers $E_F$ as evidenced by ambipolar conduction. $\mu = 371\ cm^2\ V^{-1}\ s^{-1}$, $n = 9.3 \times 10^{16}\ cm^{-3}$ for electrons. The scale bars correspond to 3 μm in **a** and 30 μm in **b**.

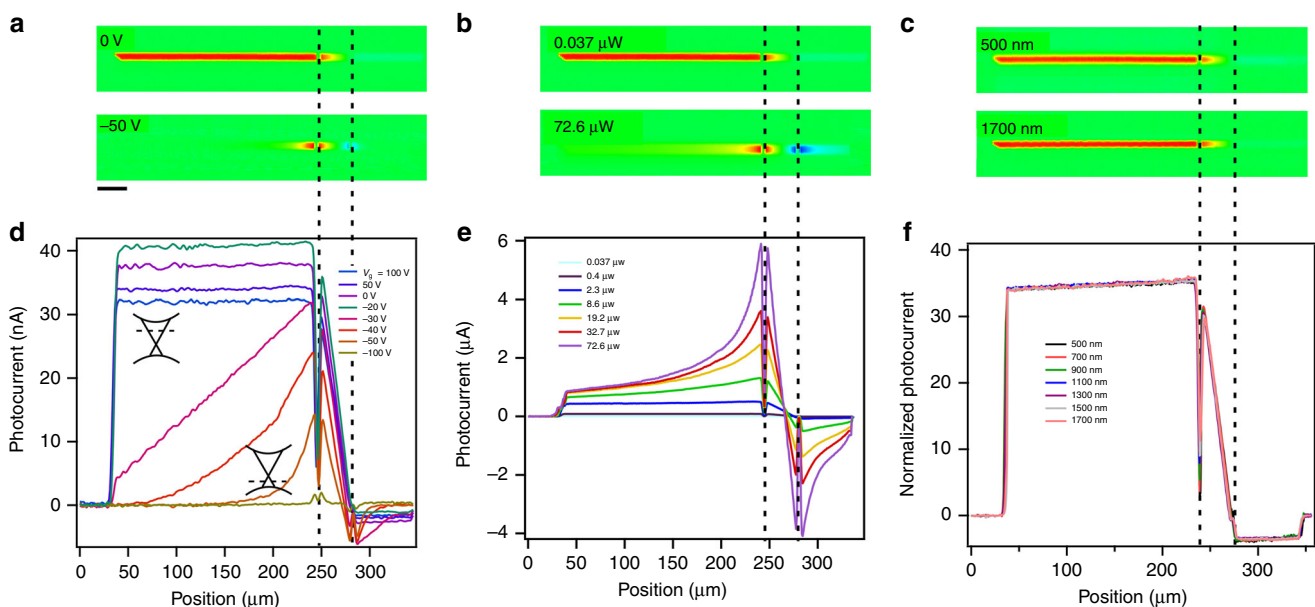

**Fig. 4 Effects of wavelength, gate voltage and laser power on photocurrent decay lengths in Sb-doped $Bi_2Se_3$.** The dashed lines indicate the contacts. The measurements are carried out at 7 K and zero source–drain and gate biases. **a–c** 2D photocurrent maps. **d–f** line cuts along the nanoribbons axis. **a, d** Gate voltage dependence. Inset, band diagrams showing $E_F$ position. Laser power is 166 nW. **b, e** Laser power dependence. **c, f** Wavelength dependence. Laser power from 77 to 280 nW was used for different wavelengths to maintain the same exciton injection rate. The scale bar denotes 30 μm.

Kosterliz–Thouless (KT) phase transition[49], which predicts the phase transition of a 2D system into a superfluid under the KT transition temperature $T_{KT}$, despite the absence of conventional long-range order in 2D. In this theory, the vortex excitations emerge and become closely bound for $T < T_{KT}$, resulting in a coherent state that displays frictionless exciton transport[49,50]. For $T > T_{KT}$, the vortices are unbound and the correlation length scales exponentially with $T$[51] $\left(\xi = c_1 \exp\left[\frac{c_2}{T - T_{KT}}\right]^{1/2}\right)$. The characteristic length of exciton transport, that is, $L_d$, is then expected to have a similar scaling and decrease exponentially at higher temperature. The $T_{KT}$ values estimated by the Hartree–Fock mean-field calculations are in good agreement with the experimental observation (Supplementary Fig. 11). Finally, the topological exciton condensate, as implied by the above observations, is unique in that it results from direct excitons at the TI surface, while excitons in previous systems are indirect with electrons and holes either spatially separated by an insulating layer[1–4,6] or at energy minima with different momenta[8]. Consequently, the coherent macroscopic quantum states are robust and can be realized in simple devices that do not involve complex structures, enabling widespread applications in quantum computations and spintronics.

## Methods

**Nanoribbon growth and device fabrication**. The CVD growth was carried out in a Lindberg Blue M tube furnace, following similar procedures as in previous work[29]. The system was first evacuated to a base pressure of 30 mTorr and Ar was then injected and a room pressure was maintained. For a typical growth, 116 mg of $Bi_2Se_3$ powder (99.999%, Alfa Aesar) was mixed with 20–35 mg of Sb powder (99.999%, Alfa Aesar) and placed in a small quartz tube at the centre of the tube furnace. Se pellets (250 mg) (99.999%, Johnson Matthey Inc.) were placed in another quartz tube upstream by a distance of 16 cm. A silicon substrate was placed 14 cm downstream from the centre of the furnace. The surface of the silicon substrate was coated with 10 nm of Au as a catalyst by electron beam evaporation. The temperature at the centre of the furnace was 680 °C, the Ar flow rate was 150 sccm (standard cubic centimetres per minute) and the growth time was 5 h. After that, the furnace was cooled down to room temperature over ~3 h. The growth yields both nanoribbons and nanoplates (Supplementary Fig. 1a). The as-grown nanoribbons were then transferred to 300 nm $SiO_2$ covered Si substrates, where single nanoribbon field-effect transistor (FET) devices were fabricated using a standard electron beam lithography process. Top metal contacts (10 nm Cr/290 nm Au or 10 nm Ti/290 nm Au) were made using an electron beam evaporator (CHA) or a sputterer (Lesker). A typical device is shown in Supplementary Fig. 1b, c.

**Optoelectronic measurements**. The low temperature measurements were performed in a cryostat (Janis ST-500). Current–voltage curves were measured through a current preamplifier (DL Instruments, model 1211) and a NI data acquisition system. SPCM measurements were performed using a home-built setup based upon an Olympus microscope. Briefly, a 532 nm CW laser or a tuneable laser (NKT SuperK plus) was focused by a ×10 NA 0.3 objective lens to a diffraction limited spot with a size of ~3 μm and raster scanned on a planar nanoribbon device by a pair of mirrors mounted on galvanometers, while both reflectance and photocurrent were simultaneously recorded to produce 2D maps. The laser power was controlled by a set of neutral density (ND) filters and was measured by a power meter underneath the objective lens. Fast photoresponse measurements were performed using a pulsed laser (Thorlabs 450 nm, pulse width 10–40 ns), high-speed amplifiers (Femto DHPCA-100) and a digital oscilloscope. The results in Figs. 1, 3 and 4 (except Fig. 3a, c) are obtained from one device for consistency, while the general trends are highly repeatable in more than 10 devices measured to date.

**Micro-ARPES**. Micro-ARPES experiments were performed at the Microscopic and Electronic Structure Observatory beamline 7.0.2 at the Advanced Light Source. Samples were removed from growth chamber, sealed in an argon gas environment and inserted into the micro-ARPES UHV end-station with a base pressure better than $5 \times 10^{-11}$ mbar via attached glovebox. ARPES data were collected at 70 K using a hemispherical Scienta R4000 electron analyser, 100 eV photon energy and a beam size of 50 μm². A typical micro-ARPES spectrum of a Sb-doped $Bi_2Se_3$ nanoplate is shown in Supplementary Fig. 2.

## Data availability

The data that support the findings of this study are available from the corresponding author upon reasonable request.

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

## Acknowledgements

This work was supported by National Science Foundation Grant DMR-1838532 and DMR-1710737. S.Y.S. was supported by National Science Foundation Grant DMR-1832728. This research used the Molecular Foundry and the Advanced Light Source, which are US Department of Energy Office of Science User Facilities under Contract No. DE-AC02-05CH11231. H.F. acknowledges the US National Science Foundation Research Experiences for Undergraduates (REU) programme under Grant No. PHY-1560482. We acknowledge Alex Weber-Bargioni, Edward Bernard and Hans Bechtel at the Molecular Foundry for the assistance on optical measurements.

## Author contributions

D.Y. and Y.H. designed the experiments. Y.H. synthesized $Bi_2Se_3$ nanoribbons and performed the measurements. R.X., L.M., H.C.T., J.F. and H.F. assisted the synthesis and measurements. A.R., I.V. and E.R. performed micro-ARPES measurements. R.W., O.E., B.W. and S.S. performed theoretical calculation. All the authors analysed the data. D.Y., Y.H., R.W., O.E. and I.V. co-wrote the paper.

## Competing interests

The authors declare no competing interests.
