## [Peer Review File · Nature Communications]

Reviewers' comments:

Reviewer #1 (Remarks to the Author):

Report on NCOMMS-19-17741-T, Yu et al.: "Millimetre-Long Transport of Excitons in Topological Insulators"

The manuscript presents scanning photocurrent measurements from a 3D topological insulator, Sb substituted Bi₂Se₃ nanoribbons, which were characterized as possessing a single Dirac cone with the chemical potential within the bulk band gap. The study found unusually long photocurrent diffusion length that survives up to almost 200K, and becomes almost nondecaying below 40K. The observation is inconsistent with the single particle diffusion model. The Authors further argue that this could be explained by the superfluid nature of Bose-Einstein condensed electron-hole bounded pairs (excitons). Overall, this could be an important and inspiring work for the community. I have a few comments and questions listed below.

1. As the observed photocurrent seems to be independent of photon energy, it's not clear if the exciton condensation is a result of optical pumping (transient effect), or a spontaneous condensation at low temperature (static ground state). Correct me if I am wrong, I guess the Authors have in mind the former case.

2. What is the role of photoexcitation in this study? To be more specific, I have a few questions concerning the details of the photoexcitation process:

a. In the manuscript, the Authors seem to suggest the excitons condense on the surface. In the Authors' model, are the photoexcited unbounded electron-hole pairs created in the bulk or on the surface? If on the surface, please explain how the quantum efficiency is not vanishingly small as most of the light is being absorbed by the bulk (the penetration depth of light in Bi₂Se₃ reaches about 10--30 nm in the visible-NIR range [Phys.Rev.B 86, 035327])? If in the bulk, then please see my comment below.

b. Assuming the electron-hole pairs are photoexcited within the bulk, and then relax towards the Fermi level which is dominated by topological surface states. Are the e-h pairs bounded (forming excitons) within the bulk, or after they relax to the surface? If within the bulk, then why would the surface states have anything to do with the observed long photocurrent diffusion length? If the excitons are formed after relaxing into the topological surface states near the Fermi level, then please explain how excitons could form from gapless Dirac surface states, as we know that excitons are typically formed across a band gap.

3. The photon energies used in this study (0.7—2.5 eV) is much larger than the energy scale of the bulk band gap/in-gap surface states (roughly 0.4 eV). The electron-hole pairs have suffered very large energy dissipation before reaching the bulk band edge. While possible, it is hard to imagine the system always favors the same low energy excitons throughout this wide range of excitation energy, instead of other relaxation channels such as phonons, plasmons (PRL 115, 257402) and magnons (PRL 119, 136802). Please clarify why the system would prefer to relax into excitons on the surface over such large energy range? I don't know any other material with this property.

4. Scanning photocurrent microscopy is a relatively unconventional tool for probing excitons as compared to photoluminescence spectroscopy. Are there other examples where this technique was successfully used for determining the exciton condensation?

In summary, although the manuscript is nicely written with helpful illustrations, I find it hard to follow mainly due to lack of explanation to the optical processes (microscopic mechanisms) leading to surface exciton formation in the reported system. While I have no doubt that the observed dissipationless photocurrent is real, I don't think the work will be impactful enough for Nature Communication unless the Authors could explain the nature and the formation of excitons in the studied material, which is the critical element for the interpretation of the data. Therefore, I would recommend the Authors take my comments into consideration and make appropriate adjustments before the publication.

Reviewer #2 (Remarks to the Author):

The manuscript by Hou et al. presented very interesting photocurrent results on 3D TI nanoribbons. The photocurrent in their Sb-doped Bi₂Se₃ nanoribbon was observed even if the laser beam spot was ~0.5 mm away from the contact with little decay. This observation is indeed quite striking. For me, I have not seen similar data from previous photocurrent measurements on graphene, TIs and Weyl semimetals.

Therefore, I would recommend the paper for publication because of the interesting data observation.

That being said, I am a bit skeptical about the claim of exciton and exciton condensation in their TI samples. First, the TI surface states are gapless. So it is difficult to imagine that strong excitons would form. Usually, like in conventional semiconductors, 2H-TMDs and bilayer graphene with out-of-plane electrical fields, the existence of excitons is shown by the observation of strong absorption peaks (or photoluminescence) at energies below the single particle band gap. By contrast, here, there is no spectroscopy observation of the formation of excitons. Without the observation of excitons as the first step, the observation of exciton condensation seems to be far-fetched.

I would agree that exciton condensation can be one plausible explanation of their data. So I would recommend the authors to focus more on the striking photocurrent data by themselves, and treating the exciton condensation as one of the possible interpretations.

1. Response to Referee #1

The manuscript presents scanning photocurrent measurements from a 3D topological insulator, Sb substituted Bi₂Se₃ nanoribbons, which were characterized as possessing a single Dirac cone with the chemical potential within the bulk band gap. The study found unusually long photocurrent diffusion length that survives up to almost 200K, and becomes almost nondecaying below 40K. The observation is inconsistent with the single particle diffusion model. The Authors further argue that this could be explained by the superfluid nature of Bose-Einstein condensed electron-hole bounded pairs (excitons). Overall, this could be an important and inspiring work for the community. I have a few comments and questions listed below.

Author's response: We thank the reviewer for confirming the potential importance of our work.

1. As the observed photocurrent seems to be independent of photon energy, it's not clear if the exciton condensation is a result of optical pumping (transient effect), or a spontaneous condensation at low temperature (static ground state). Correct me if I am wrong, I guess the Authors have in mind the former case.

Author's response: Yes, we indeed believe that the exciton formation and exciton condensation are due to optical pumping and the resulted excitons are in a transient state. To make this point more clear, **we added "non-equilibrium" before excitons in the abstract.**

2. What is the role of photoexcitation in this study? To be more specific, I have a few questions concerning the details of the photoexcitation process:

a. In the manuscript, the Authors seem to suggest the excitons condense on the surface. In the Authors' model, are the photoexcited unbounded electron-hole pairs created in the bulk or on the surface? If on the

surface, please explain how the quantum efficiency is not vanishingly small as most of the light is being absorbed by the bulk (the penetration depth of light in Bi₂Se₃ reaches about 10--30 nm in the visible-NIR range [Phys. Rev. B 86, 035327])? If in the bulk, then please see my comment below.

Author's response: The role of photoexcitation in this study is to generate electrons and holes, which we believe form excitons at low temperature. According to previous theoretical works [reference 10 (Phys. Rev. B 95, 205410 (2017)) and 11 (Phys. Rev. B 97, 075109 (2018)) in the manuscript], exciton condensates are expected to form at the surface of the TIs. We believe that the photon absorption mainly happens in the bulk because of the optical penetration depth is much larger than a quintuple layer thickness as the reviewer pointed out. The photogenerated carriers then relax into the surface states and form excitons at the surface. To clarify, **we added a sentence on page 8 of the revised manuscript, "the photoexcitation mainly generate charge carriers in the bulk of the TI material."**

b. Assuming the electron-hole pairs are photoexcited within the bulk, and then relax towards the Fermi level which is dominated by topological surface states. Are the e-h pairs bounded (forming excitons) within the bulk, or after they relax to the surface? If within the bulk, then why would the surface states have anything to do with the observed long photocurrent diffusion length?

Author's response: We don't think the excitons form in the bulk. Instead, we believe that the carriers photogenerated mainly in the bulk relax towards the topological surface states and form excitons at the TI surface. This carrier relaxation process from bulk to the TI surface has been observed by time-resolved ARPES in intrinsic TI and has a timescale of a few picoseconds [reference 15 (Phys. Rev. B 98, 115132 (2018)) and 16 (Sci. Rep. 7, 14080 (2017)) in the manuscript]. Subsequent relaxation of carriers in the surface states is slow and could be over hundreds of picoseconds [reference 15 and 16]. Excitons are expected to form after reaching the surface where electrons and holes have much longer lifetime. Exciton formation in an inversely occupied Dirac cone at the TI surface has recently been theoretically predicted by Phys. Rev. B 95, 205410 (2017), Phys. Rev. B 97, 075109 (2018) and Nat. Commun. 10, 210 (2019).

If the excitons are formed after relaxing into the topological surface states near the Fermi level, then please explain how excitons could form from gapless Dirac surface states, as we know that excitons are typically formed across a band gap.

Author's response: In a gapped system with bandgap E_g , excitons are located in the gap with energy $E_g - E_b$, where E_b is the binding energy (Figure R1a, adapted from Science 358, 907-910 (2017)). As long as carriers are long-lived, the strong attractive interaction between electrons and holes can generate a bound state of excitons even in a gapless system. Exciton formation in gapless Dirac surface states is predicted to open two many-body energy gaps at the quasi Fermi levels of electrons and holes respectively, as shown in Fig. R1c (adapted from Phys. Rev. B 95, 205410 (2017)). This gap opening is analogous to the energy gap associated with Cooper pairs in superconductors. The excitons are stable since its corresponding state with exciton gaps enjoys a lower energy compared to the original gapless state. This mechanism for exciton formation in gapless systems has been studied in topological insulators (Phys. Rev. Lett. 103, 066402(2009); Phys. Rev. B 95, 205410 (2017)), and Weyl semimetals (Phys. Rev. Lett. 109, 196403 (2012)).

[Redacted]

Figure R1. Energy diagrams for a gapped system (a) and TIs (b-c). a, Excitonic states in bilayer graphene. Adapted from Science 358, 907, 2017. b-c, TI surface states without (b) and with excitonic gap (c). Adapted from Phys. Rev. B 95, 205410 (2017).

To clarify on the exciton formation process, we have added a few sentences on page 8:

"Then these photoexcited carriers undergo a fast relaxation process within a few ps and relax to the surface states where recombination is much slower up to hundreds of ps. The electrons and holes at the surface form excitons at sufficiently low temperature. The exciton formation opens many-body energy gaps at the surface states similar to the energy gap associated with Cooper pairs in superconductors^{10,11}."

3. The photon energies used in this study (0.7–2.5 eV) is much larger than the energy scale of the bulk band gap/in-gap surface states (roughly 0.4 eV). The electron-hole pairs have suffered very large energy dissipation before reaching the bulk band edge. While possible, it is hard to imagine the system always favors the same low energy excitons throughout this wide range of excitation energy, instead of other relaxation channels such as phonons, plasmons (PRL 115, 257402) and magnons (PRL 119, 136802). Please clarify why the system would prefer to relax into excitons on the surface over such large energy range? I don't know any other material with this property.

Author's response: We thank the reviewer for the insightful questions. The photoexcited electron relaxation process is complicated and follows several steps as shown in Fig. R2. We think in our system the excess energy of the photoexcited electrons is mostly released into phonons, but we do not exclude the possibility of plasmons and magnons as the reviewer suggests. The relaxation of carriers from high energy states to bulk band edge and surface states is within a few picoseconds as demonstrated by previous time-resolved ARPES results (reference 14-18 in the manuscript). On the other hand, the carriers at the surface states enjoy a lifetime up to hundreds of picoseconds, where they form low-energy excitons. Though the photon energy used in our experiment 0.7-2.5 eV is larger than the bulk bandgap 0.3 eV limited by our tunable laser bandwidth, most of these carriers are believed to relax to surface states because the slow carrier relaxation at the surface becomes the bottleneck of the entire relaxation process. This slow relaxation at the surface states is indeed unusual and surprising, but confirmed by the ARPES data and may be attributed to the small density of states at the Dirac point.

Our experimental results including the mm-long photocurrent decay lengths are best understood with the exciton model. We have considered other mechanisms including plasmons but they unlikely account for our experimental results as detailed below.

[Redacted]

Figure R2. Schematics of carrier dynamics over the transition energy range, adapted from reference PRL 108, 117403 (2012).

Plasmons: Surface plasmons were observed in PRL 115, 257402 by a MEELS study. In our study, surface plasmons are unlikely to explain the data because:

a) Optical pumping is used to excite the device at normal incidence in our setup. Thus there is no direct coupling between photons and surface plasmons because of momentum mismatch. In order to have efficient plasmon excitation, structured surfaces, such as metallic gratings (*Phys Rev Lett* 1968; 21: 1530–1533), nanoslits (*Opt Express* 2005; 13: 6815–6820) and permittivity gradient (*Light: Science & Applications* volume 5, page e16179 (2016)), are required to compensate for the mismatched momentum of SPPs and free-space light. On the other hand, our nanoplate surface is flat and not structured.

b) As we described in the manuscript, surface plasmon dies out as it loses energy due to absorption. Thus its propagation length depends on the imaginary part of the surface refractive index which is wavelength dependent (*reference 40 (NPG Asia Mater.* 9, e425 (2017)) and 41 (*Surface Plasmon Resonance Based Sensors. (Springer, Berlin, Heidelberg, 2006))* in the manuscript). This is in contrary to the observed wavelength independent photocurrent decay length in our study.

c) Surface plasmons are expected to exist at high electron density as shown in PRL 115, 257402, but we only observed long photocurrent decay lengths in intrinsic samples.

Magnons: a) Magnons in PRL 119, 136802 is a result of excitation to the second Dirac cone located 1.5 eV above the conduction band edge. In our study, photocurrent is observed down to 0.7 eV photon energy, which indicates that second Dirac cone is not involved.

b) The observed chiral spin mode in this paper sensitively depends on the circular and linear polarization of the light. On the other hand, our photocurrent distribution under normal incidence configuration does not depend on the polarization of the laser.

We cited both PRL 115, 257402 and PRL 119, 136802 in the revised manuscript. We have also added further explanation to exclude the possibility that photocurrent is generated by plasmon or magnon.

On pages 8, we added:

"The normal incidence configuration with light injection from free space also unlikely creates surface plasmon due to momentum mismatch. Surface plasmons are expected to exist at high electron density, but we only observed long photocurrent decay lengths in nearly intrinsic samples."

On page 22, we added:

"Magnons in reference 54 is a result of excitation to the second Dirac cone located 1.5 eV above the conduction band edge. In our study, photocurrent is observed down to 0.7 eV photon energy, which indicates that second Dirac cone is not involved. In addition, the observed chiral spin mode in reference 54 sensitively depends on the circular and linear polarization of the light. On the other hand, our photocurrent distribution under normal incidence configuration does not depend on the polarization of the laser."

4. Scanning photocurrent microscopy is a relatively unconventional tool for probing excitons as compared to photoluminescence spectroscopy. Are there other examples where this technique was successfully used for determining the exciton condensation?

Author's response: As far as we know, scanning photocurrent microscopy (SPCM) has not been used to determine the exciton condensation anywhere else. On the other hand, a similar technique, spatially resolved photoluminescence (PL), has been used as evidence of exciton condensation and superfluidity in double quantum wells (reference 2,3,5 in the manuscript). In these studies, PL images exhibit macroscopically ordered patterns, or PL peak intensity sharply increases with reduced peak widths at lower temperature. Bi₂Se₃ does not show strong light emission and hence PL imaging is not a suitable method for studying excitons in this material. SPCM is a powerful experimental technique that can be applied to visualise the transport of locally photoexcited charge carriers. Compared to spatially resolved PL, it does not require materials to have strong light emission and is hence ideal to study TIs. In our study, the observed mm-long photocurrent decay length with nanosecond lifetime indicates that excitons propagate highly dissipationlessly, which can be understood by formation of exciton condensate. We have compared SPCM to PL imaging on page 3.

In summary, although the manuscript is nicely written with helpful illustrations, I find it hard to follow mainly due to lack of explanation to the optical processes (microscopic mechanisms) leading to surface exciton formation in the reported system. While I have no doubt that the observed dissipationless photocurrent is real, I don't think the work will be impactful enough for Nature Communication unless the Authors could explain the nature and the formation of excitons in the studied material, which is the critical element for the interpretation of the data. Therefore, I would recommend the Authors take my comments into consideration and make appropriate adjustments before the publication

Author's response: We thank the reviewer for the valuable suggestion. We agree with the reviewer that the carrier relaxation process was not clearly explained in the original manuscript. **We have revised our manuscript in several places as indicated in the detailed response above**, to clarify the optical processes leading to surface excitons, and to discuss the plasmon and magnon process, following the reviewer's suggestion.

2. Response to Referee #2

The manuscript by Hou et al. presented very interesting photocurrent results on 3D TI nanoribbons. The photocurrent in their Sb-doped Bi₂Se₃ nanoribbon was observed even if the laser beam spot was ~0.5 mm away from the contact with little decay. This observation is indeed quite striking. For me, I have not seen similar data from previous photocurrent measurements on graphene, TIs and Weyl semimetals.

Therefore, I would recommend the paper for publication because of the interesting data observation.

That being said, I am a bit skeptical about the claim of exciton and exciton condensation in their TI samples. First, the TI surface states are gapless. So it is difficult to imagine that strong excitons would form. Usually, like in conventional semiconductors, 2H-TMDs and bilayer graphene with out-of-plane electrical fields, the existence of excitons is shown by the observation of strong absorption peaks (or photoluminescence) at energies below the single particle band gap. By contrast, here, there is no spectroscopy observation of the formation of excitons. Without the observation of excitons as the first step, the observation of exciton condensation seems to be far-fetched.

I would agree that exciton condensation can be one plausible explanation of their data. So I would recommend the authors to focus more on the striking photocurrent data by themselves, and treating the exciton condensation as one of the possible interpretations.

Author's response: We thank the reviewer for the positive comments. In the manuscript, we have carefully considered and excluded other mechanisms that can lead to photocurrent in TI. Excitonic superfluid is so far the only reasonable way we identified to explain all experimental observations reported in the manuscript. However, we agree with the reviewer that we cannot exclude the possibility that a new theory may be developed in the future that explains the results better. Following this suggestion, we have stated clearly in the revised manuscript that exciton condensation is only one of the possible interpretations.

On page 8, we changed

"We attribute the observed highly efficient carrier transport at low temperature to formation of superfluid-like exciton condensate in TIs."

to

"One possible way to understand the observed highly efficient carrier transport at low temperature is the formation of superfluid-like exciton condensate in TIs."

Reviewers' comments:

Reviewer #1 (Remarks to the Author):

The Authors have clarified all my questions and concerns in their Reply. The experimental observation of dissipationless photocurrent on Bi₂Se₃ is surprising and very interesting. Although in my opinion, the existence of excitons is still debatable without direct experimental support such as PL or optical absorption data. The result of a large nonlocal photocurrent being generated in a wide frequency range is concrete and interesting. Regardless of the origin of the effect being excitonic superfluid or not, this could be an important discovery for applications with the photovoltaic cells and optoelectronics.

As the Authors pointed out on page 10, the present study also poses interesting implications on the debate of whether ODLRO could exist in an excitonic insulator or not [Ref. 45 vs Ref.46, see also references Phys. Rev. B 11 3317 and Phys. Rev. B 90 195118]. However, one should note that an insulating state is not the opposite of a superfluid phase. The superfluidity in Ref.46 is referring to the "excitation energy", not the flow of mass or charge as in the superfluid phase of He₄ or superconductors. Therefore, it is not straight forward to have a dissipationless nonlocal photocurrent even in the presence of excitonic condensation (I don't see how it is done). I think it is important to distinguish these differences (i.e. superfluidity in He₄ vs excitonic condensation) and explain in the manuscript how the superflow of energy leads to a dissipationless current.

In conclusion, I would recommend publishing this manuscript on Nature Communication with minor modifications.

Reviewer #2 (Remarks to the Author):

I would like to thank the authors for the replies.

I have read carefully the replies to my questions and those from the other reviewer.

Again, my overall evaluation is that the data observations are striking and have not been seen in similar experiments on other systems before. "Exciton condensation" could be an explanation. However, I am still concerned the indirect nature of the photocurrent experiments: There is no even direct signature of exciton yet, let alone a condensation. All arguments are based on photocurrent transport behaviours). Another example is that, by reading the replies to the other reviewer, one can see several places the authors used the phrase : "We believe ...". For instance, the reviewer asked whether the excitons are formed by bulk or surface states. The authors said that "they believe" that excitons are formed by the surface states, but the authors didn't provide direct observation to support their believe. This, again, shows that there are many microscopic processes that are not directly probed by the photocurrents. Therefore, the authors have to make assumptions.

By bringing this up, I am not saying that one cannot make assumptions. Again, I believe that exciton condensation is one explanation. But, given the level of assumptions that have to be made, I think the authors should scale down their claim further. If that is done properly, I would be very happy to support publication. My advice is focusing more on the striking experimental results by themselves and less about the exciton condensation.

Just to give a few examples, the title is still : "Millimetre-Long Transport of Excitons in Topological 1 Insulators".

The abstract: "we report experimental evidence of formation and efficient transport of non-equilibrium excitons in $\text{Bi}_{2-x}\text{Sb}_x\text{Se}_3$ nanoribbons... We confirmed that photoexcited electrons and holes are paired into bound states ...". The first sentence is fine. It says "evidence" indicating possible but not definitive. But the second sentence says "confirmed" meaning that the authors are sure that there are excitons in their photoexcited nanoribbons. Overall, the tone of the manuscript is not consistent.

1. Response to Referee #1

The Authors have clarified all my questions and concerns in their Reply. The experimental observation of dissipationless photocurrent on Bi₂Se₃ is surprising and very interesting. Although in my opinion, the existence of excitons is still debatable without direct experimental support such as PL or optical absorption data. The result of a large nonlocal photocurrent being generated in a wide frequency range is concrete and interesting. Regardless of the origin of the effect being excitonic superfluid or not, this could be an important discovery for applications with the photovoltaic cells and optoelectronics.

Author's response: We thank the reviewer for the positive comments. We agree with the referee that our explanation of the experimental results by excitons is plausible but is still debatable. We have softened the claims by emphasizing that exciton condensation is a possible mechanism instead of being completely confirmed. The exact changes we made can be seen in our response to the second referee.

As the Authors pointed out on page 10, the present study also poses interesting implications on the debate of whether ODLRO could exist in an excitonic insulator or not [Ref. 45 vs Ref.46, see also references Phys. Rev. B 11 3317 and Phys. Rev. B 90 195118]. However, one should note that an insulating state is not the opposite of a superfluid phase. The superfluidity in Ref.46 is referring to the “excitation energy”, not the flow of mass or charge as in the superfluid phase of He₄ or superconductors. Therefore, it is not straight forward to have a dissipationless nonlocal photocurrent even in the presence of excitonic condensation (I don't see how it is done). I think it is important to distinguish these differences (i.e. superfluidity in He₄ vs excitonic condensation) and explain in the manuscript how the superflow of energy leads to a dissipationless current.

Author's response: We thank the reviewer for this insightful question. We agree with the reviewer that the nature of an excitonic condensate has been controversial. Experimentally, signatures of both excitonic superfluid (Ref 3, 4, 6, 7) and excitonic insulator (Ref 12) have been reported. The super-flow of an exciton condensate and its electrical detection was discussed (*Nature Physics volume4, pages799–802*

(2008)) and demonstrated by Coulomb drag measurements in double layer quantum hall systems (Ref 4 in the manuscript: *Nature* 488, 481-484 (2012)) and double bilayer graphene (Ref 6 in the manuscript: *Nat. Phys.* 13, 746-750 (2017)).

Theoretically, there is no consensus on whether an excitonic condensate state is an insulator or it can sustain a superflow of excitons. Ref. 45 pointed out that the off-diagonal long-range order (ODLRO), which is present in He⁴ superfluid, is absent in a condensed exciton system. However, Ref. 46 (Ref 48 in the revised manuscript) showed that although the ODLRO in the particle-particle channel does not exist, the ODLRO in the particle-hole channel does exist in condensed excitonic state. Superfluidity of the exciton condensate is generally understood in the framework of Kosterlitz-Thouless (KT) transition (Ref. 49 in the revised manuscript: *J. Phys. C: Solid State Phys.* 6, 1181 (1973)). Phys. Rev. B 90, 195118 (the reference that this referee mentioned) predicts that the insulating states are favored when phonons are involved.

More specifically, as pointed out by the referee, the excitonic superfluidity in Ref. 46 (Ref 48 in the revised manuscript) refers to the “excitation energy”, not the flow of mass or charge as in the superfluid phase of He⁴ or superconductors. We note here it is not necessary to have a flow of mass or charge in order to create non-local photocurrent as observed in our experiment. As long as the excitation energy created by local laser injection flows to the electrical contact, excitons are effectively generated near the contact. Then as we stated on page 8, the separation of these excitons under strong electric field near the metal junction results in a photocurrent.

To make this point clear, we added more discussions on page 8:

“The flow of excitons does not generate an electrical current because excitons are charge neutral. However, as excitons reach the metal-TI contact, they are separated and create photocurrent.”

We also cited *Phys. Rev. B* 11 3317 and *Phys. Rev. B* 90 195118 in the revised manuscript.

We added on page 10 the following sentences to clarify more on the controversy over whether an exciton condensate is a insulator or superfluid.

“Experimentally, signatures of both excitonic insulator and excitonic superfluid have been reported.”

We also added a few sentences on page 10 to distinguish the differences between superfluidity in He⁴ and exciton condensate and explain how the superflow of energy leads to dissipationless current, as the referee suggested:

"As pointed out in reference 48, superfluidity in He⁴ or superconductors can be distinct from that in exciton condensate, where the former is via mass flow and the later is via energy flow. It is interesting to note that mass flow is not necessary for the observed long photocurrent decay. The energy flow from the photoexcitation point to electrical contact can also result in non-local photocurrent."

In conclusion, I would recommend publishing this manuscript on Nature Communication with minor modifications.

2. Response to Referee #2

I would like to thank the authors for the replies.

I have read carefully the replies to my questions and those from the other reviewer.

Again, my overall evaluation is that the data observations are striking and have not been seen in similar experiments on other systems before. "Exciton condensation" could be an explanation. However, I am still concerned the indirect nature of the photocurrent experiments: There is no even direct signature of exciton yet, let alone a condensation. All arguments are based on photocurrent transport behaviours). Another example is that, by reading the replies to the other reviewer, one can see several places the authors used the phrase : "We believe ...". For instance, the reviewer asked whether the excitons are formed by bulk or surface states. The authors said that "they believe" that excitons are formed by the surface states, but the authors didn't provide direct observation to support their believe. This, again, shows that there are many microscopic processes that are not directly probed by the photocurrents. Therefore, the authors have to make assumptions.

By bringing this up, I am not saying that one cannot make assumptions. Again, I believe that exciton condensation is one explanation. But, given the level of assumptions that have to be made, I think the authors should scale down their claim further. If that is done properly, I would be very happy to support publication. My advice is focusing more on the striking experimental results by themselves and less about the exciton condensation.

Just to give a few examples, the title is still: "Millimetre-Long Transport of Excitons in Topological 1 Insulators".

The abstract: "we report experimental evidence of formation and efficient transport of non-equilibrium excitons in Bi_{2-x}SbxSe₃ nanoribbons... We confirmed that photoexcited electrons and holes are paired into bound states ..." The first sentence is fine. It says "evidence" indicating possible but not definitive. But the second sentence says "confirmed" meaning that the authors are sure that there are excitons in their photoexcited nanoribbons. Overall, the tone of the manuscript is not consistent.

Author's response: We thank the reviewer for confirming the novelty and importance of the work. To make the tone consistent, we have made the following modifications:

We changed the title from

“Millimetre-Long Transport of **Excitons** in Topological Insulators”

to

"Millimetre-Long Transport of **Photogenerated Carriers** in Topological Insulators"

On page 2, we changed the sentence

“We confirmed that photoexcited electrons and holes are paired into bound states by electric field independent photocurrent distributions.”

to

“The photocurrent distributions are independent of electric field, indicating that photoexcited electrons and holes form excitons.”

On page 4, we changed the sentence

“Here we apply scanning photocurrent microscopy (SPCM) in intrinsic 3D TIs to **demonstrate** the formation of excitons and their transport of macroscopic distance.”

to

“Here we apply scanning photocurrent microscopy (SPCM) in intrinsic 3D TIs **to provide evidence on** the formation of excitons and their transport of macroscopic distance.”

On page 10, we changed the sentence

"The observed highly dissipationless **exciton transport** in TIs provides strong evidence supporting superfluidity."

to

"The observed highly dissipationless **transport of photogenerated carriers** in TIs provides strong evidence supporting superfluidity."

We believe that the changes and additions we made in the revised manuscript and the answers provided above address all of the questions raised by the reviewers. We hope that the revised manuscript is now acceptable for publication in *Nature Communications*. Please do not hesitate to contact us if you have any further questions. We sincerely appreciate your consideration of our manuscript.